# Train Trajectory-Following Control Method Using Virtual Sensors

**DOI:** 10.3390/s24165385

**Published:** 2024-08-20

**Authors:** Youpei Huang, Xiaoguang Ma, Lihui Ren

**Affiliations:** 1Rail Transit Institute, Tongji University, Shanghai 200333, Chinapz12621@163.com (X.M.); 2CRRC Nanjing Puzhen Co., Ltd., Nanjing 210031, China

**Keywords:** articulated vehicle, lateral control, nonlinear dynamics, trajectory following

## Abstract

Trajectory-following control is the basis for the practical application of an articulated virtual rail train transportation system. In this paper, a planar nonlinear dynamics model of an articulated vehicle is derived using the Euler–Lagrange method. A trajectory-following control strategy based on the first following point is proposed, and a feedback linearization control algorithm is designed based on the vehicle dynamics model to achieve the trajectory following of the rear vehicle. Based on the target trajectory formed by the first following point and measured by virtual sensors, a vector analysis method grounded in geometric relationships is proposed to solve in real time for the desired position, velocity, and acceleration of the vehicle. Finally, a MATLAB/SIMPACK dynamics virtual prototype is established to test the vehicle’s trajectory-following effectiveness and dynamics performance under lane change and circular curve routes. The results indicate that the control algorithm can achieve trajectory following while maintaining good vehicle dynamics performance. It is robust to variations in vehicle mass, vehicle speed, tire cornering stiffness, and road friction coefficient.

## 1. Introduction

With the rapid growth of worldwide urbanization in China, traffic congestion on urban roads has become a major problem that is hindering the comprehensive development of cities. Developing public transportation is one of the most effective ways to solve urban traffic congestion. A virtual rail transit system, as a new type of urban transportation system, effectively fills the capacity gap between bus system and subway [1,2]. A short construction period, low cost, flexible operation, environmental protection, and aesthetic advantages make it a popular choice for enterprises in the rail transportation industry and local governments. With the rise of virtual rail train-related companies and their engineering applications, the academic research and engineering applications related to virtual rail trains are gradually becoming hotspots for universities and research institutes [3,4,5].

Virtual rail trains must operate in a similar manner to road vehicles; for example, they must be able to stay in a fixed lane without interfering with traffic in other lanes. However, when a multi-group articulated train passes through a curve, even if the tractor is able to stay in the original lane by the driver’s skillful operation, the trailer’s trajectory is often shifted inward, resulting in the trailer not being able to stay in the original lane [6,7]. As a kind of articulated train with multiple formations, the virtual rail train also has the above-mentioned disadvantages. In order to ensure that the overall vehicle can stay on the original track, it is a feasible solution to realize the track following of the trailer for the tractor. All axles of the virtual rail train can be steered independently to provide the conditions for trajectory following [8,9].

Research has been conducted on trajectory following of articulated trains by many scholars. T-Kaneko et al. [10,11,12] presented a steering control system for achieving zero-departure tracking of an articulated passenger car. The control system consists of a feedforward controller and a feedback controller, and the controller is verified to achieve good lane-tracking performance through a simulation and an experiment. Sebastian Wagner et al. [13,14,15] proposed a 2-degree-of-freedom multi-articulated train with an all-wheel steering control scheme, which consists of a feedforward controller and a feedback controller. The nonlinear feedback controller compensates for the uncertainty and disturbance of the model and tracks the motion trajectory mechanically and stably. Islam M M et al. [16,17] designed a rear axle active steering control strategy using optimal pre-targeting control and an LQR optimization algorithm based on a simplified dynamics model of a heavy semi-trailer. Meanwhile, unified optimization indexes for low and high speeds were designed, and the vehicle structure parameters, driver model parameters, and control strategy parameters were optimized by using a genetic algorithm. The simulation shows that the rear axle active control strategy can achieve low-speed path-following control and high-speed stability control for the semi-trailer. Leng Han et al. [18,19,20] studied the trajectory-following control problem of articulated trains with gantry-type architecture, and designed a control algorithm by combining the optimal pre-sighting control method with the trajectory-following control strategy of articulated trains. In order to improve the control effect, model prediction and control time delay were included in the control algorithm. Sun Bangcheng [21,22] investigated the trajectory-following control of virtual rail trains with gated architectures, developed a Liapunov function-based control algorithm for the vehicle, and finally implemented the vehicle guidance control by using differential steering. In order to avoid the influence of modeling uncertainty and external disturbance on the track-following control of the vehicle, Zhang Zonghua [23] and Esmaeili N [24] designed an adaptive sliding mode controller to improve the control robustness of the vehicle, and verified the effectiveness of the control algorithm by building an ADAMS virtual prototype and a TRUCKSIM virtual prototype. Yuan Xiewen et al. [9,25,26] introduced the sensing subsystem and control subsystem of a virtual rail train trajectory-following system, studied the lane marking recognition effect in dark and daytime by real vehicle tests, and proposed two trajectory control algorithms applicable to tram: the pre-sight PID algorithm and the model predictive control algorithm. The results of the real-vehicle test for the preview PID algorithm show the precise stopping and trajectory control of the virtual rail train.

The current axle-steering control method uses passive control, such as setting a fixed proportional relationship between the steering angles of the axles, and while the front vehicle is steering, the rear axle is steering according to the set proportional coefficient [6]. This method can effectively reduce the overall vehicle turning radius and the trajectory deviation of the rear vehicle, and improve the maneuverability of the vehicle. Also, to further reduce the trajectory deviation, S. MANESIS [27] designed the articulated device as a movable connection, releasing the lateral motion coupling between the front and rear carriage. The passive control strategy is simple to implement, inexpensive, and reliable, so it is often used in the early articulated trains dealing with rear axle steering problems. The disadvantage of this method, however, is that the vehicle has difficulty adapting to trajectory paths with changing curvature, e.g., vehicle lane changes and transition curve routes where straight lines enter a circular line. To allow the vehicle to pass through the curve mentioned above, designers often widen the lane as much as possible, while requiring it to slow down as much as possible, in order to reduce the danger caused by over-width and lateral offset as the vehicle passes the line. For the active control scheme, the trajectory-following control strategy needs to give the vehicle trajectory in advance, and then control the vehicle to run along the established trajectory [28]. This control strategy can allow the vehicle to operate autonomously, but it is difficult to predict its trajectory in advance in the complex operating environment of urban roads. Determining the trajectory by the trajectory planning algorithm often increases the computational cost and complexity of the control system. In contrast, having a human manually drive the vehicle to select a trajectory suitable for the current moment is a historically proven direct and feasible method.

Although the virtual rail train is essentially multi-articulated vehicles, its operating environment is different and the vehicle structure is more diverse. So far, virtual rail train structures under research include three-module six-axles [29], four-module six-axles [30], four-module five-axles [18,19,20], three-module eight-axles [31], seven-module eight-axles [23,32], and five-module ten-axles [33] structures. All the above research objectives are the study of vehicle self-steering with full-axle steering, aiming to realize the vehicle’s following of the existing trajectory through axle steering control without considering a human driver in operation. At the same time, the above research content lacks the discussion on the characteristics of vehicle dynamics, such as the treatment of the control variables greater than the model state for the structure of three-module six-axles, four-module eight-axles, four-module six-axles, and five-module ten-axles models. In this paper, based on a certain type of four-module six-axles virtual rail train, the trajectory-following control of the above scenario is studied, and a control method for trajectory following of the rear vehicles is proposed. In this method, the first steering wheel is controlled by the driver to form the target trajectory, and then the control algorithm is tasked with controlling the remaining steering axle to follow the rear vehicle control point for the target trajectory. The control algorithm first derives a planar nonlinear dynamics model of the articulated vehicle based on the Euler–Lagrange method, and then designs the control law of the vehicle dynamic system using the feedback linearization method. Since the number of control variables of the vehicle system is more than the number of states of the vehicle, which makes the solution of the control variables lack conditions, the coordination equation for redundant control variables is constructed using the principle of minimum wear. Since the control law is a nonlinear equation about the control variables, Newton’s iterative method is used to solve for the control variables. The target trajectory of the rear following point is generated and recorded by the operation of the first following point in real time. In order to transform the target trajectory into the desired state of the vehicle, a vector solution method based on geometric relations is proposed to obtain the desired position of the vehicle by real-time solution, and then the desired velocity and desired acceleration of the vehicle are solved according to the vehicle motion model and trajectory information. Finally, the virtual prototype built by MATLAB/SIMPACK is used to test the effectiveness of the vehicle control algorithm in the lane change and circular curve routes. This paper provides a reference for an articulated virtual rail train with a redundant steering axle structure.

## 2. Virtual Rail Train Dynamics Model

The virtual rail train consists of four carriages and six steerable axles, and each axle has two tires. The vehicle is actually composed of two articulated semi-trailers connected in reverse formation. There are two axles under the end module (MC1, MC2) and one axle bridge under the middle module (TP1, TP2). In the xy plane, the carriages are connected to each other by an articulation rotating around the *Z*-axis, and the suspension structure adopts a double-wishbone independent suspension. Refer to Appendix A for the notational definitions of the derivation of the vehicle model. The planar motion model of the vehicle can be simplified to the single-track model shown in Figure 1. In this model, a local coordinate system is fixed at the center of mass of each vehicle body, and the *x*-axis of the coordinate system coincides with the vehicle and the *y*-axis is perpendicular to the vehicle.

This planar motion model ignores the pitch and roll motion of the vehicle. Assuming that the carriages are rigidly connected by hinges, the constraints between the carriages are geometrically constrained, and therefore, the vehicle planar motion is a holonomic constraint system. Considering the articulation as a revolute, each articulation reduces the number of degrees of freedom of the vehicle by 2. The number of degrees of freedom of each carriage in plane motion is 3, which are translations in the x and y directions and rotation in the z direction. Therefore, the number of degrees of freedom in the planar vehicle model is 6, as calculated by Equation (1):(1)$$N_{DOF}=3n-2(n-1)=n+2$$
where n is the number of carriages, which is 4, and n−1 is the number of articulations.

Since the vehicle system has 6 degrees of freedom, six independent variables are required for the description of the vehicle motion. Taking the position of the center of mass of the first vehicle and the carriage heading angle of each vehicle as independent variables, the plane position and attitude of the vehicle can be described completely, as shown in Figure 2, for the geometric meaning of the variables:(2)$${q=(x,y,\varphi_{1},\varphi_{2},\varphi_{3,}\varphi_{4})}^{T}$$
where x, y are the horizontal and vertical coordinates of the first carriage (MC1) coordinate frame, $\varphi_{i}$ is the heading angle of the *i*-th carriage.

The equation of motion of the vehicle based on the variable q can be obtained from the Euler–Lagrange equation as follows:(3)$$\frac{d}{dt}(\frac{\partial E_{K}}{\partial \dot{q}}{)}^{T}-(\frac{\partial E_{K}}{\partial q}{)}^{T}=Q$$
where $E_{K}$ is the kinetic energy of the vehicle and Q is the generalized force vector of the system. The system kinetic energy is as follows:(4)$$E_{K}=\frac{1}{2}\sum m_{i}(v_{ix}^{2}+v_{iy}^{2})+I_{i}\omega_{i}^{2}$$
where $m_{i},I_{i}$ are the mass and the rotational inertia around the center of mass of the *i*-th carriage, respectively. $v_{ix},v_{iy,}\omega_{i}$ are the translational velocity and angular velocity of vehicle coordinate frame i with respect to the inertial coordinate frame.

The carriages’ center-of-mass velocity is expressed as follows:(5)vixviy=d(LKi:ηx(q))dtd(LKi:ηy(q))dt

Among them:$$L_{K}=\left(\begin{array}{ccccc} 1 & 0 & 0 & 0 & 0\\ 1 & l_{G1} & l_{K2} & 0 & 0\\ 1 & l_{G1} & l_{G2}+l_{K2} & l_{K3} & 0\\ 1 & l_{G1} & l_{G2}+l_{K2} & l_{G3}+l_{K3} & l_{K4} \end{array}\right)$$
$$\eta_{x}(q)={\left(\begin{array}{ccccc} x & -\cos\varphi_{1} & -\cos\varphi_{2} & \cdots & -\cos \end{array}\varphi_{n}\right)}^{T}$$
$$\eta_{y}(q)={\left(\begin{array}{ccccc} y & -\sin\varphi_{1} & -\sin\varphi_{2} & \cdots & -\sin \end{array}\varphi_{n}\right)}^{T}$$

The angular velocity of the system is expressed as follows:(6)$$\omega_{i}={\dot{\varphi }}_{i}$$

For the generalized force vector, according to the principle of virtual work we have the following:(7)$$Q=2\sum_{i=1}^{j}\frac{\partial r_{A_{i}}^{T}}{\partial q}F_{A_{i}}$$
where j is the number of axles of the vehicle; $r_{A_{i}}={\left(x_{A_{i}},y_{A_{i}}\right)}^{T}$ is the position vector of the *i*-th axle, $F_{A_{i}}$ is the force vector acting on the *i*-th axle, and the number ‘2’ indicates the number of tires per axle.

For the position vector of the axles, there are the following:(8)rAi=(LAi:ηx(q),LAi:ηy(q),)

Among them:$$L_{A}=\left(\begin{array}{ccccc} 1 & -l_{A2} & 0 & 0 & 0\\ 1 & l_{A2} & 0 & 0 & 0\\ 1 & l_{G1} & l_{K2}+l_{A3} & 0 & 0\\ 1 & l_{G1} & l_{K2}+l_{G2} & l_{K2}-l_{A4} & 0\\ 1 & l_{G1} & l_{K2}+l_{G2} & l_{K3}+l_{G3} & l_{K4}-l_{A5}\\ 1 & l_{G1} & l_{K2}+l_{G2} & l_{K3}+l_{G3} & l_{K4}+l_{A6} \end{array}\right)$$

For $F_{A_{i}}$, there are the following:(9)FAi=RIotκjRκjotAi(fixAi,fiyAi)
where RκjotAi=cos(γi)−sin(γi)sin(γi)cos(γi) and fixAi,fiyAi are the longitudinal and lateral forces of the tire, respectively. The lateral force is calculated from the tire model, which is presented below. The longitudinal forces are the control variables to the system. $\gamma_{i}$ is the steering angle of the *i*-th axle, which is the control variable of the system, as shown in Figure 2.

For the calculation of lateral forces, using the tire model shown in Figure 3, there are the following:(10)fiyAi=c(γi−βi)
where βi=tan−1(vAiyκjvAixκj) represents the description of the velocity direction of the *i*-th axle $A_{i}$ in the coordinate frame $\kappa_{j}$.

Based on Equations (3) to (10), the dynamic differential equation for the vehicle’s planar motion is obtained:(11)$$M(q)\ddot{q}+G(q,\dot{q})=J_{x}(\gamma ,q)F_{x}+J_{y}(\gamma ,q)(\gamma -\beta (q,\dot{q}))$$
where M(q) is the mass matrix, $G(q,\dot{q})$ describes the forces generated by the Centripetal and Coriolis forces
Fx=(f1xA1,f2xA2,…,f6xA6),γ=(γ1,γ2,…γ6), β=(β1,β2,…β6),Jx(γ)=(J1:1,J2:1,…,J6:1),Jy(γ)=(J1:2,J2:2,…,J6:2),Ji=2∑∂rAiT∂qRIotκjRκjotAi.

## 3. Controller Design

The vehicle planar dynamics model derived in the previous section is used for the controller design of vehicle trajectory following. It is assumed that the vehicle motion $\left(q,\dot{q}\right)$, driver information $F_{x}$, $\gamma_{0}$, and the control point $p_{1}$ motion $\left({\ddot{x}}_{p1},{\ddot{y}}_{p1},{\dot{x}}_{p1},{\dot{y}}_{p1},x_{p1},y_{p1}\right)$ can be measured.

A general overview of the control strategy in this paper is depicted in Figure 3. The vehicle moves under the driver’s maneuver, and the trajectory of the first following point is recorded and used to calculate the desired motion of the vehicle. The controller of the vehicle is divided into two parts, the main controller and the redundant axles controller. The redundant axles controller calculates the pure rolling state of the axles based on the vehicle’s motion and outputs the steering angle to the main controller and the test vehicle. The main controller solves the steering angle of the remaining axles for the test vehicle using the control algorithm designed based on the planar motion model derived in the previous section. The information of the vehicle’s actual motion, desired motion, the driver’s input, and the output of the redundant controller is needed in the main controller.

### 3.1. Control Strategy

According to the proposed control strategy, we only need to control four control points of the vehicle to realize the following of the trajectory. And the deviation of the four following points from the trajectory can be translated into the deviation of the vehicle heading angle state from the desired state, i.e.,:(12)$$e=q_{er}-q_{e}$$

Here:$$\left\{\begin{array}{l} q_{er}={\left(\varphi_{1r},\varphi_{2r},\varphi_{3r},\varphi_{4r}\right)}^{T}\\ q_{e}={\left(\varphi_{1},\varphi_{2},\varphi_{3},\varphi_{4}\right)}^{T} \end{array}\right.$$

For descriptive convenience, let:(13)$$\left\{\begin{array}{l} q_{dr}={\left(x_{r},y_{r}\right)}^{T}\\ q_{d}={\left(x,y\right)}^{T} \end{array}\right.$$

Thus, Equation (11) can be re-expressed as follows:(14)$$\begin{array}{l} \left(\begin{array}{cc} M_{dd} & M_{de}\\ M_{ed} & M_{ee} \end{array}\right)\left(\begin{array}{c} {\ddot{q}}_{d}\\ {\ddot{q}}_{e} \end{array}\right)+\left(\begin{array}{c} G_{12}\\ G_{36} \end{array}\right)=\left(\begin{array}{cc} J_{dd}^{x} & J_{de}^{x}\\ J_{ed}^{x} & J_{ee}^{x} \end{array}\right)\left(\begin{array}{c} F_{12}^{x}\\ F_{36}^{x} \end{array}\right)\\ +\left(\begin{array}{cc} J_{dd}^{y} & J_{de}^{y}\\ J_{ed}^{y} & J_{ee}^{y} \end{array}\right)\left[\left(\begin{array}{c} \gamma_{12}\\ \gamma_{36} \end{array}\right)-\left(\begin{array}{c} \beta_{12}\\ \beta_{36} \end{array}\right)\right]+D(t) \end{array}$$

Here, D(t) denotes the disturbance term of the system, which contains the uncertainty of the vehicle model and external noise, etc. $M=\left(\begin{array}{cc} M_{dd} & M_{de}\\ M_{ed} & M_{ee} \end{array}\right)$, $J_{x}=\left(\begin{array}{cc} J_{dd}^{x} & J_{de}^{x}\\ J_{ed}^{x} & J_{ee}^{x} \end{array}\right)$, $J_{y}=\left(\begin{array}{cc} J_{dd}^{y} & J_{de}^{y}\\ J_{ed}^{y} & J_{ee}^{y} \end{array}\right)$; G12=G1−2:G36=G3−6:, γ12=γ1−2:γ36=γ3−6:, F12x=Fx1−2:F36x=Fx3−6:, β12=β1−2:β36=β3−6:.

Define the control algorithm of the system as follows:(15)$$\begin{array}{l} J_{ed}^{x}F_{12}^{x}+J_{ee}^{x}F_{36}^{x}+J_{ed}^{y}(\gamma_{12}-\beta_{12})+J_{ee}^{y}(\gamma_{36}-\beta_{36})=\\ M_{ed}{\ddot{q}}_{d}+G_{36}+M_{ee}({\ddot{q}}_{er}+K_{d}\dot{e}+K_{p}e+K_{i}\int e) \end{array}$$

Substituting the above equation into Equation (14), since the first control point formed the target trajectory, the linear closed-loop system equation of motion can be obtained by removing the first two rows of the equation as
(16)$$\dddot{z}+K_{d}\ddot{z}+K_{p}\dot{z}+K_{i}z=d$$
where $\dot{z}=e$ denotes the generalized displacement error, and $d=M_{ee}^{-1}D$ denotes the model uncertainty and signal noise.

When d is 0 or bounded constant, according to the Routh’s stability criterion, the closed-loop system converges if $\left\{\begin{array}{l} K_{d}K_{p}-K_{i}>0\\ K_{d},K_{p},K_{i}>0 \end{array}\right.$. Since the components of the model uncertainty are unknown, in order to obtain a better control effect, the uncertainty of the system parameters should be as accurate as possible to prevent d from being too large, resulting in system divergence.

### 3.2. Redundant Control Variables

In the control law of Equation (14), the number of control quantities, $\gamma_{2},\gamma_{3},\ldots ,\gamma_{6}$, to be solved is five, but there are only four constraint equations, so it is obviously not possible to solve for all the control quantities. One way is to solve the remaining control quantities by measuring one of the control quantities to be solved, $\gamma_{c}$, and then substituting it into Equation (14). Therefore, a separate control law needs to be designed for $\gamma_{c}$. Using the principle of minimizing the tire wear of $\gamma_{4}$, the control law is designed as in Equation (17). The control law is such that the lateral force of $\gamma_{4}$ is zero.
(17)$$\gamma_{4}=\beta_{4}$$

### 3.3. Solution Calculation

It can be seen from Equation (11) that $J_{y}\gamma$ contains the nonlinear components $\gamma_{i}\cos\gamma_{i},\gamma_{i}\sin\gamma_{i}$. The curves of the functions f(x)=xcosx, f(x)=xsinx are shown in Figure 4. The functions are in the interval of [−pi/2, pi/2], and the independent variable and the dependent variable are not a single mapping relationship, and there are multiple solutions in the process of solving the function’s inverse, which needs to be judged and excluded. In general, the tire steering motion is continuous, so the difference in solution, Δγ=γ(k)−γ(k−1), should be as small as possible to make the results smooth and continuous.

In order to calculate the control inputs $\gamma_{2},\gamma_{3},\gamma_{5},\gamma_{6}$, subtract the left side by the right side of Formula (15) and define it as h(γ) in Equation (18), and the nonlinear equation h(γ)=0 is obtained. By solving the nonlinear equations, the control inputs can be obtained.
(18)$$\begin{array}{l} h(\gamma )=J_{ed}^{x}F_{12}^{x}+J_{ee}^{x}F_{36}^{x}+J_{ed}^{y}(\gamma_{12}-\beta_{12})+J_{ee}^{y}(\gamma_{36}-\beta_{36})-\\ M_{ed}{\ddot{q}}_{d}+G_{36}+M_{ee}({\ddot{q}}_{er}+K_{d}\dot{e}+K_{p}e+K_{i}\int e) \end{array}$$

In Equation (18), there are four equations and four control variables to be solved. The matrices such as M and J, which are dependent on $q,\dot{q}$, have been derived in Equation (11). The longitudinal vector F is zero, except for the first element f0x, which is the driving force to maintain vehicle cruise control since the rolling resistance is small and has little effect on the lateral motion of the vehicle. β and G are formulas of $q,\dot{q}$, also derived in Equation (11). From the above discussion, Equation (18) contains factors such as $\gamma_{i}\cos\gamma_{i},\gamma_{i}\sin\gamma_{i}$, and the equations have strong nonlinear characteristics. Methods for solving nonlinear equations need to be used.

By using the Newton–Raphson method, there are the following:(19)$$\gamma (k+1)=\gamma (k)-{\left.h(\gamma (k))(\frac{\partial h}{\partial \gamma }{)}^{-1}\right|}_{\gamma (k)}$$

${\left.(\frac{\partial h}{\partial \gamma }{)}^{-1}\right|}_{\gamma (k)}$ is the Jacobian matrix of h(γ). When ${\left\|\gamma (k+1)-\gamma (k)\right\|}_{2}<\varepsilon$, (ε is the solution accuracy), γ(k+1) is the numerical solution of the equations. The maximum number of iterations needs to be set in order to prevent too many iterations and increase the computation time. According to Figure 4, it can be seen that Equation (18) has multiple solutions, and the results obtained by the Newton iterative method should be judged by the results. Since the steering motion of the vehicle is continuous, the solutions obtained from two solutions should be close to each other. For example, (0.217, 0.2) and (1.428, 0.2) in Figure 4 are both solutions of f(x)=0.2, but the difference between the transverse coordinates of the two points is large. To ensure the continuity of the control variables, the solution closest to the value of the initial point should be chosen as the final solution.

According to the iterative computation test, after the initial point is determined, the convergence result is likely to converge to the solution near the initial point when the iteration step is small. Therefore, in the process of solving, the initial point of the solution can be set to the value of the vehicle’s steering angle at the previous moment, in order to ensure the convergence result near the current state value with a higher probability, which reduces the calculation volume of the solution and improves the calculation efficiency.

The pseudocode for the control algorithm introduced above is shown in Algorithm 1.
**Algorithm 1: Pseudocode for control algorithm**1**Initializes** the vehicle state and existing historical trajectory;2**Loop**3**Input**: driving and first axle steering angle;4Vehicle model state measurement for q and $\dot{q}$;5Redundant axle control $\gamma_{4}$ calculation using Equation (17);6Historical trajectories are integrated and updated as target path:
$\left(x_{p},y_{p}\right)$;7Transform the target trajectory into the target state:
$q_{er},{\dot{q}}_{er},{\ddot{q}}_{er}$;8Solve Equation (18) for the control quantity
$\gamma_{2},\gamma_{3},\gamma_{5},\gamma_{6}$ using Newton–Raphson method;9**Output:** $\gamma_{2},\gamma_{3},\gamma_{5},\gamma_{6}$ to vehicle prototype;10**Exit loop**11Operation result observation and analysis.

## 4. Virtual Prototype Application

### 4.1. Virtual Prototype of Vehicle Model

In order to test the feasibility of the control algorithm and its control effectiveness, virtual simulation testing was used. Based on MATLAB/SIMPACK software, a four-module, six-axles virtual rail train simulation prototype shown in Figure 1 was built. The simulation prototype contains four carriages, six steerable suspensions, three vehicle articulation devices, and 12 tires. The steering mechanism of the suspension is simplified to a four-link mechanism. The tire model uses the Magic Formula empirical model. The number of degrees of freedom of motion for the entire vehicle is 60, and the vehicle model parameters are referred to in Appendix B. The output signals of the simulation prototype are the longitudinal force and the steering angle of the first axle, the vehicle motion state, and the position, speed, and acceleration of the first control point. The input signals of the simulation prototype are the steering angles of the remaining five steering axles. On the contrary, the output signal and the input signal of the simulation prototype are the input and output signals of the control system, respectively. The simulation model of the vehicle + control system is shown in Figure 5.

### 4.2. Controller Model

For ride comfort consideration, the cut-off frequency of the closed-loop system of the vehicle plus control system should be below 1.2 Hz.

The transfer function is as follows:(20)$$e(s)=\frac{s}{s^{3}+K_{d}s^{2}+K_{p}s+K_{i}}d(s)=\frac{s}{{\left(s+\lambda \right)}^{3}}d(s)$$

Then, the characteristic root of the system can be obtained as s=−λ. For a positive real number λ, the system deviation is convergent to the response of the disturbance and the system is stable. Considering the cut-off frequency and response bandwidth of the above system, we take λ = 5, which corresponds to the system Bode diagram in Figure 6. According to the diagram, the cut-off frequency of the system is 6.83 rad/s (1.08 Hz < 1.2 Hz) and the response bandwidth is 5.27 rad/s (0.84 Hz), which meets the requirements of vehicle design. At the same time, according to Figure 6, it can be seen that the system suppression gain of deviation is −36 dB, located at the frequency of 3.65 rad/s (0.58 Hz). The control system parameters now are $K_{d}=15,K_{p}=75,K_{i}=125$.

### 4.3. Simulation Results and Analysis

This section simulates a vehicle passing through two types of curves at a speed of 30 km/h: a circular curve and a lane change curve with a width of d = 3.5 m. The geometry of the line is shown in Figure 7. The vehicle first travels a distance in a straight line, then enters the circular curve and the lane change curve, after which it returns to travel in a straight line. For a circular curve line, the curvature of the curve is not continuous at the connection of the straight line and the circular line. The sudden change in curvature causes the sudden change in lateral acceleration of the vehicle, which is not conducive to the comfort of vehicle operation. Therefore, in order to make the curvature continuous, a fifth polynomial curve with small distance is spliced at the connection between the straight line and the circular line. The reason for using the fifth-order polynomial is that the left side of the polynomial curve has three boundary conditions, namely, the original function is continuous, the first-order derivative is continuous and the second-order derivative is continuous (the curvature corresponds to the second-order derivative of the curve), and three parameters are required. Similarly for the right-hand side, there are the same three boundary conditions which also correspond to three parameters. Thus, six parameters correspond to a fifth-order polynomial curve.

The simulation results of the circular curve route are shown in Figure 8. The first carriage enters the circular curve at s = 100 m, t = 8.2 s, and the last carriage leaves the circular curve at s = 182 m, t = 22.6 s. Figure 8a shows the results of the trajectory deviation of the control point. From the figure, it can be seen that the trajectory deviation is mainly generated in the transition between entering and leaving the curve, the maximum is 5 cm, and the trajectory deviation decays to 0 quickly in the circular curve. This result shows that the control algorithm achieves trajectory following well. Figure 8b,c show the yaw rate profile of the vehicle and the lateral acceleration profile of the vehicle’s center of mass, respectively. From the plots, it can be seen that a small amount of overshoot, 7% and 9%, respectively, is generated during the transition of the vehicle from a straight line to a circular curve, which leads to the trajectory deviation of the vehicle’s curve negotiation. Figure 8d shows the steering angle of the vehicle. From the figure, it can be seen that the steering curve is relatively smooth, and there is no rapid change in direction behavior. At the same time, it can be seen from Figure 8c that the vehicle is traveling with a lateral acceleration close to 0.25 g. This value of lateral acceleration is near the upper limit of the design criteria of the vehicle, which is a tough driving condition. However, the dynamic performance observed in Figure 8a,c,d shows that the vehicle operates with a good dynamic performance. Table 1 shows the control effects with different parameters, which indicates that the control method is well worked in different road conditions.

In order to demonstrate the superiority of the proposed control algorithm, the simulation results of the existing passive control methods are compared. The existing control method is applied to SRT, and the control strategy is that the steering angle of the rear axle is 1/2 of the adjacent hinged angle. The control method adopts the geometric steering principle based on Ackermann, ignores the side deflection characteristics of the tire, and considers that the vehicle’s hinge point can run stably in the same circular trajectory under the action of the hinge angle and the axle steering angle. For the application of the Ackermann steering principle in articulated trains, please refer to [19] for extended Ackermann steering geometry. The simulation results of the above circular curve using this passive control method are shown in Figure 9. As can be seen from the figure, with the passive control method, the maximum trace following error when entering and leaving the circular curve is 1.38 m, and the error on the circular curve is slightly reduced to 0.79 m. Obviously, in the above working conditions, the lateral acceleration of the vehicle is large, and the tire needs to provide a large side force, so ignoring the side dynamic characteristics tends to produce a large control error. The above comparison results show that the new control algorithm, which takes into account the tire’s sideward characteristics and trajectory information feedback, has obvious advantages over the passive control in existing engineering applications.

## 5. Conclusions

The proposed trajectory-following control strategy for a four-module six-axles virtual rail train demonstrates significant effectiveness. The control algorithm, derived using the Euler–Lagrange method and designed with feedback linearization, ensures that the rear vehicle accurately follows the target trajectory formed by the first steering point. This method effectively manages various vehicle parameters, maintaining robust performance despite changes in vehicle mass, speed, cornering stiffness, and road friction coefficients. The results from the virtual prototype tests conducted in MATLAB/SIMPACK validate the algorithm’s capability to handle lane changes and circular curve routes, achieving a maximum following error of less than 0.1 m across different speeds and curves.

This study’s findings highlight that the control strategy not only achieves precise trajectory following but also maintains good dynamic performance. The innovative approach of controlling redundant steering axles based on minimizing tire wear, combined with Newton’s iterative method for solving nonlinear control equations, proves to be efficient. The real-time solution of desired vehicle states, derived from geometric relationships and vehicle motion models, further enhances the control system’s effectiveness. The algorithm’s robustness to vehicle parameter variations is a crucial feature, ensuring consistent performance under diverse operating conditions.

Overall, this research provides a comprehensive and practical solution for the trajectory-following control of articulated virtual rail trains. The integration of virtual sensors for real-time measurement and the use of a detailed planar nonlinear dynamics model offer a solid foundation for further advancements in urban rail transit systems. The successful application of the control algorithm in a virtual prototype sets a precedent for future implementations and optimizations in real-world scenarios.

Although the control algorithm proposed in this paper can be well applied to this type of train, the control algorithm needs a detailed prior understanding of the vehicle physical model and its related parameters to reduce the following deviation caused by model uncertainty. In the follow-up research, a model-free control (MFC) method based on data-driven will be sought to further improve the robustness of the control system to model uncertainties [29,30,31,32,33,34].

## Figures and Tables

**Figure 1 sensors-24-05385-f001:**
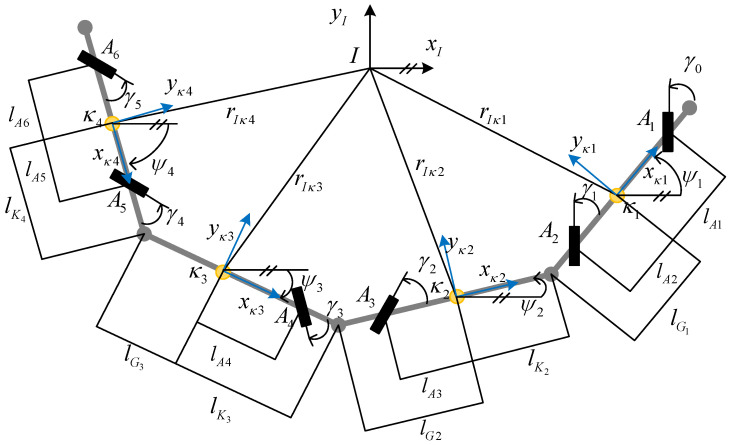
Virtual rail train planar kinematic model.

**Figure 2 sensors-24-05385-f002:**
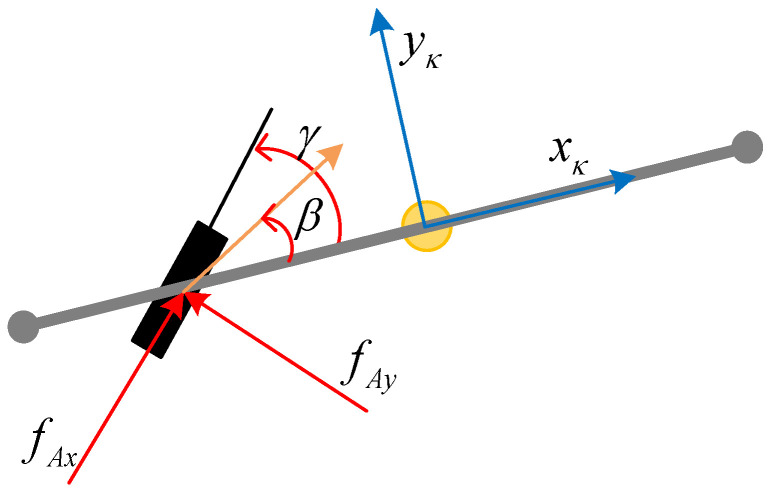
Tire model.

**Figure 3 sensors-24-05385-f003:**
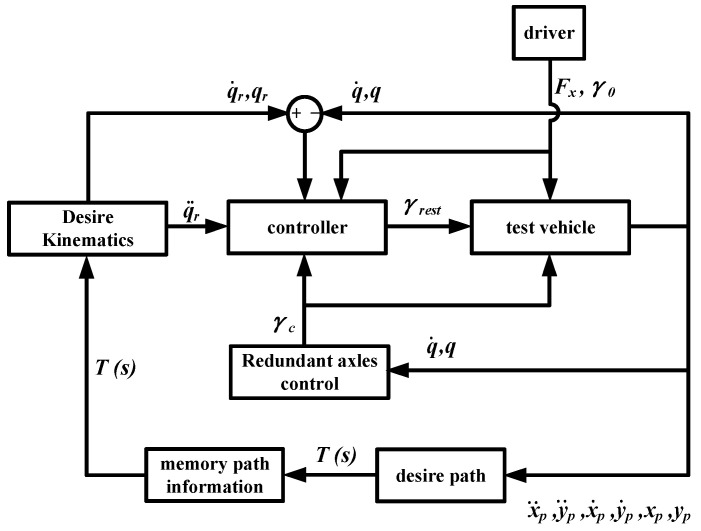
Control strategy block diagram.

**Figure 4 sensors-24-05385-f004:**
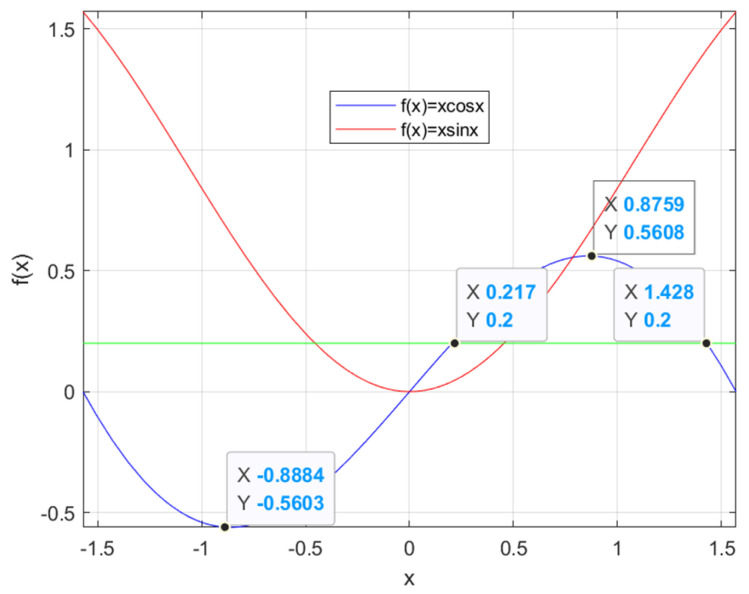
Inverse solving of f(x)=xcosx and f(x)=xsinx.

**Figure 5 sensors-24-05385-f005:**
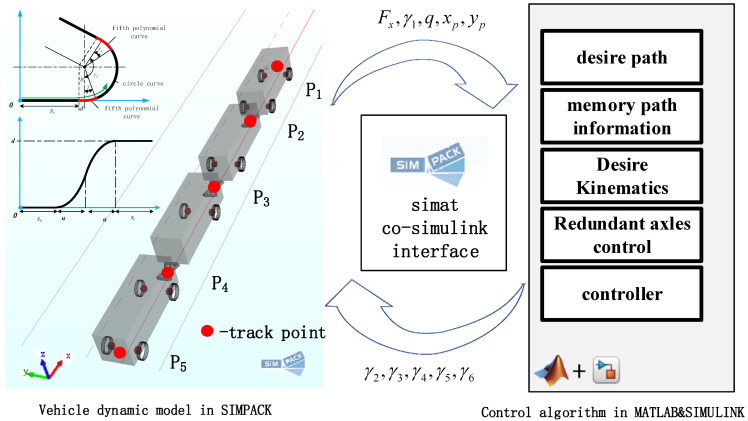
Vehicle + control simulation model in MATLAB/SIMPACK.

**Figure 6 sensors-24-05385-f006:**
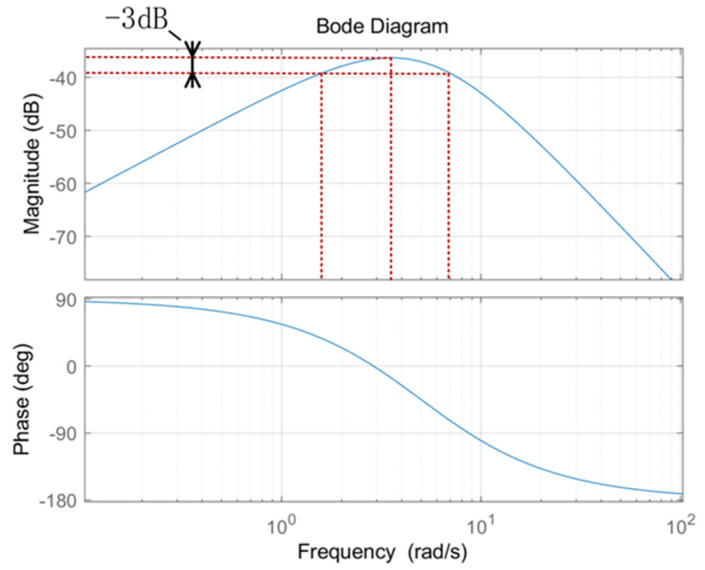
Bode plot of disturbance transfer function.

**Figure 7 sensors-24-05385-f007:**
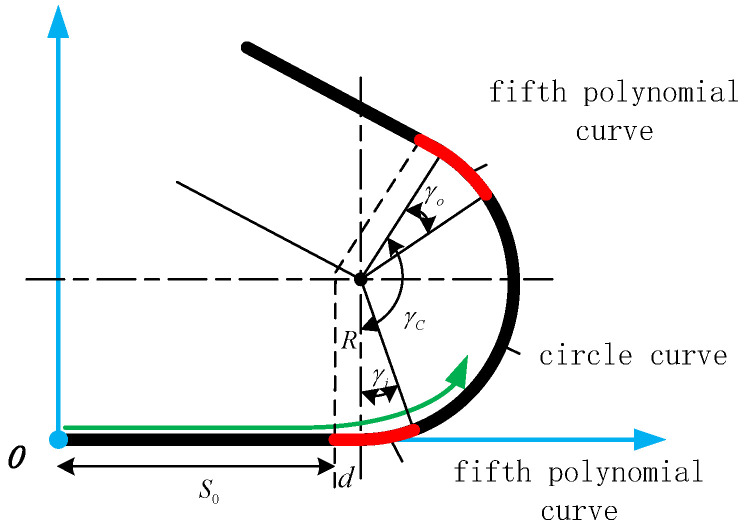
Circular line.

**Figure 8 sensors-24-05385-f008:**
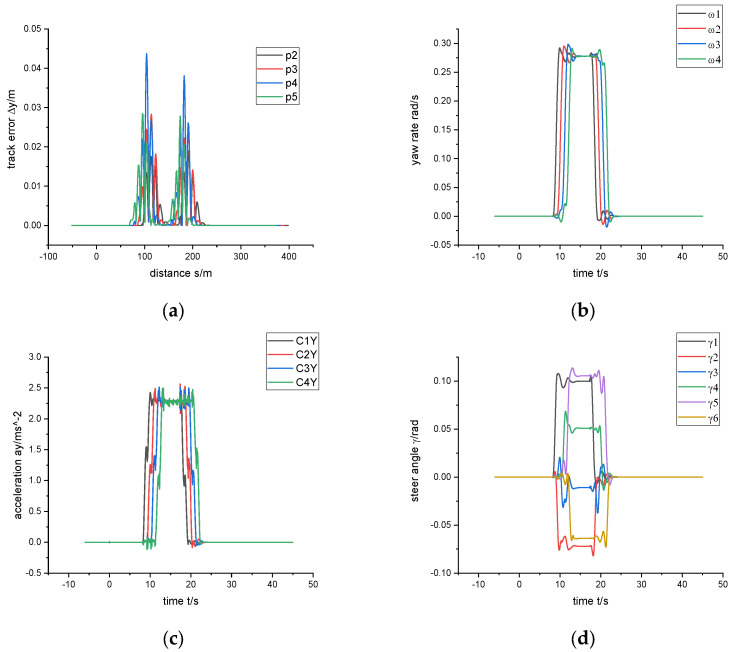
Speed 30 km/h through R30m circular curve (**a**) trajectory error, (**b**) yaw rate, (**c**) lateral acceleration of center of mass, (**d**) steering angle.

**Figure 9 sensors-24-05385-f009:**
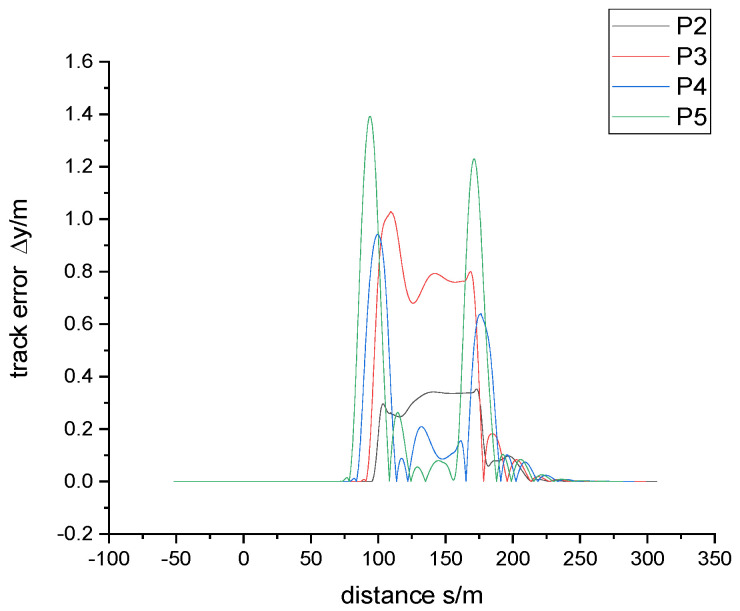
Track error in simulation condition of speed 30 km/h through R30m circular curve with existing passive control.

**Table 1 sensors-24-05385-t001:** Control effects with different parameters.

Line	Parameters	Calculation Results		
		Lateral Acceleration	Overshoot	Track Error
Circular curve	R = 25, v = 7 m/s	0.2 g	10%	5 cm
	R = 20, v = 5 m/s	0.13 g	8%	5 cm

## Data Availability

No new data were created or analyzed in this study. Data sharing is not applicable to this article.

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
