# Peer review of "Train Trajectory-Following Control Method Using Virtual Sensors"

_sensors, 2024, doi:10.3390/s24165385_

Round 1

Reviewer 1 Report

Comments and Suggestions for Authors

In this paper, a planar nonlinear dynamics model of an articulated vehicle is derived using the Euler-Lagrange method. The authors propose a trajectory following control strategy based on the first following point and design a feedback linearization control algorithm rooted in the vehicle dynamics model to achieve trajectory following for the rear vehicle. Utilizing the target trajectory formed by the first following point and measured by virtual sensors, a vector analysis method based on geometric relationships is proposed to solve in real time for the desired position, velocity, and acceleration of the vehicle. Finally, a MATLAB/SIMPACK dynamics virtual prototype is established to test the vehicle's trajectory following effectiveness and dynamic performance under lane change and circular curve routes. The results demonstrate that the control algorithm can achieve trajectory following while maintaining good vehicle dynamics performance. It is robust against variations in vehicle mass, vehicle speed, tire cornering stiffness, and road friction coefficient.

Overall, this is an interesting paper. The paper has a clear organization and flows well. However, the following comments shall be addressed before the paper can be considered publishable. 

1. It is not clear where equation 1 comes from. Is it based on the mobility formula? This should be clarified. 

2. Does the control-oriented model in (11) consider damping/frictional effect?

3. (11) looks like a very standard Euler-Lagrange robotic system, so the reviewer is unsure why derivations are needed in this context.

4. The control law is based on feedback linearization. However, what if there are model uncertainties (e.g. parametric uncertainties)? In such cases, the feedback linearization won't make the system exactly linear and the linear stability analysis based on the Roth-Hurwitz criterion may not hold. This should be analyzed and discussed clearly.

5. Following the previous point, it seems like the model-free control from https://ieeexplore.ieee.org/abstract/document/9714714 may offer an alternative solution to the control problem of this paper (this model-free method is robust against model uncertainties). The authors can investigate such a model-free method for future studies (this can be mentioned in the conclusion). 

6. (19) should be referred to as Newton-Raphson. 

7. Gain and phase margins should be labeled in Figure 6. 

8. There is a minor typo: after (16), the "Rouse" should be "Routh". 

Comments on the Quality of English Language

There are quite some grammar errors. Please carefully proofread the manuscript and make corrections. 

Reviewer 2 Report

Comments and Suggestions for Authors

This paper develops a trajectory following control strategy for an articulated virtual rail train system. It derives a planar nonlinear dynamics model using the Euler-Lagrange method and proposes a feedback linearization control algorithm based on this model. The approach utilizes virtual sensors to measure the target trajectory from the first following point and applies a real-time vector analysis method for determining the vehicle's desired position, velocity, and acceleration. Simulation results using MATLAB/SIMPACK show effective trajectory following and robust dynamic performance across different vehicle and road conditions. The topic is interesting, but there are the areas of improvement:

Please explain why your solution is better than the existing one?

Please cite latest papers related to the topic.

Please offer the algorithm or pseudo code of the offered solution.

Don’t use future tense in the paper, for example “The vehicle planar dynamics model derived in the previous section will be used for the controller design of vehicle trajectory following”

Please offer the detailed explanation of formula (18)

Author Response

Please see the attachment."

Round 2

Reviewer 1 Report

Comments and Suggestions for Authors

Additional comments to address:

1) A citation should be provided to equation (1).

2) The authors state, "Because the damping and friction effects are difficult to measure in practice, they are ignored in modeling and considered as uncertainty or error terms of the model." However, if these effects are treated as uncertainties, an analysis should be provided to demonstrate that this uncertainty will not degrade the stability margins of the closed-loop system and will only result in bounded tracking errors.

3) For feedback linearization, the authors need to demonstrate that the nonlinear error due to inexact cancellation is Lipschitz continuous. In such cases, the linear feedback part may stabilize the system.

4) Since the authors acknowledge that model-free control, as discussed in "https://ieeexplore.ieee.org/abstract/document/9714714," may provide an alternative solution to the control problem addressed in this paper, it would be beneficial to cite this reference. This will allow readers to explore the alternative approach in greater detail.

5) Gain and phase margins can be determined by referring to standard linear control textbooks. Specifically, if the amplitude-frequency curve does not cross 1 (0 dB) and the phase-frequency curve does not cross -180 degrees, the system may have infinite margins. Please verify this information by consulting the relevant textbooks for accuracy.

Author Response

Dear reviewer,
Thank you for your valuable feedback on our manuscript. We have carefully considered each of your suggestions and made corresponding revisions to the manuscript. The content of the reply is attached.
